# Multi-Tissue Transcriptome Analysis Identifies Key Sexual Development-Related Genes of the Ornate Spiny Lobster (*Panulirus ornatus*)

**DOI:** 10.3390/genes11101150

**Published:** 2020-09-29

**Authors:** Tomer Ventura, Jennifer C. Chandler, Tuan V. Nguyen, Cameron J. Hyde, Abigail Elizur, Quinn P. Fitzgibbon, Gregory G. Smith

**Affiliations:** 1GeneCology Research Centre, School of Science and Engineering, University of the Sunshine Coast (USC), 4 Locked Bag, Maroochydore, QLD 4558, Australia; jennifer.chandler@research.usc.edu.au (J.C.C.); tuan.nguyen@agriculture.vic.gov.au (T.V.N.); chyde1@usc.edu.au (C.J.H.); aelizur@usc.edu.au (A.E.); 2Developmental Biology and Cancer Programme, UCL Great Ormond Street Institute of Child Health, London WC1N 1EH, UK; 3Agriculture Victoria, AgriBio, Centre for AgriBiosciences, Bundoora, VIC 3083, Australia; 4Institute for Marine and Antarctic Studies (IMAS), University of Tasmania, Private Bag 49, Hobart, TAS 7001, Australia; quinn.fitzgibbon@utas.edu.au (Q.P.F.); gregory.smith@utas.edu.au (G.G.S.)

**Keywords:** sexual development, decapod crustaceans, transcriptome, sex-linked master sex regulator, doublesex and MAB-3 related transcription factors

## Abstract

Sexual development involves the successive and overlapping processes of sex determination, sexual differentiation, and ultimately sexual maturation, enabling animals to reproduce. This provides a mechanism for enriched genetic variation which enables populations to withstand ever-changing environments, selecting for adapted individuals and driving speciation. The molecular mechanisms of sexual development display a bewildering diversity, even in closely related taxa. Many sex determination mechanisms across animals include the key family of “doublesex- and male abnormal3-related transcription factors” (Dmrts). In a few exceptional species, a single Dmrt residing on a sex chromosome acts as the master sex regulator. In this study, we provide compelling evidence for this model of sex determination in the ornate spiny lobster *Panulius ornatus*, concurrent with recent reports in the eastern spiny lobster *Sagmariasus verreauxi*. Using a multi-tissue transcriptomic database established for *P. ornatus*, we screened for the key factors associated with sexual development (by homology search and using previous knowledge of these factors from related species), providing an in-depth understanding of sexual development in decapods. Further research has the potential to close significant gaps in our understanding of reproductive development in this ecologically and commercially significant order.

## 1. Introduction

Sexual development of many gonochoristic species (with discrete male and female morphs) initiates with either an environmental or a genetic cue, which sets in motion the sex determination cascade [1,2]. Although well studied in several model species, the mechanism is often perplexing and elusive with a remarkable variation in the genes and pathways involved, even between closely related species [3,4]—a great paradox, considering the shared outcome of gonochorism. Following sex determination, sexual differentiation ensues, translating the sex determining cascade into the discernible difference between the sexes. During sexual differentiation, the gonads (either testes or ovaries) develop, alongside many other sexually dimorphic traits and behavioral patterns. The gradual process of sexual differentiation leads to reproductive maturation, enabling the species to produce viable gametes, reproduce, and pass novel combinations of genes on to the next generation.

Over the past decade, transcriptomic libraries have identified a wealth of genes exhibiting sexually dimorphic expression across many animal species. In decapod crustaceans, a refined list of genes related to sexual development was elucidated, with some that have clear roles and some that are yet to be defined [5,6,7,8,9,10,11,12,13,14,15]. While the sex determination cascade is highly variable, a widely conserved element has been the family of “doublesex- and male abnormal (MAB)-3-related transcription factors” (Dmrts). The Dmrts are abundantly present across the animal kingdom, from the basal cnidarians to higher vertebrates [16]. These Dmrts all share a DM domain—a zinc finger motif that binds specific DNA sequences—thereby either activating or inhibiting transcription of genes at the vicinity of the specific DNA site. Adjacent to the DM domain, a transactivation domain (TAD) is either present or absent on the Dmrt proteins. Upon Dmrt binding to the responsive DNA element, the TAD stabilizes the dimerization of the Dmrts, which is crucial for transcriptional activation. In many species, Dmrts are differentially expressed between males and females, both with and without the presence of the TAD. This differential expression can either stabilize or destabilize the Dmrts binding to target elements in specific tissues (depending on the TAD presence or absence), thereby leading to a tissue-specific differential cascade of gene activation [3,7,17].

The regulation of Dmrts is highly divergent across ecdysozoa. In all holometabolous but only some hemimetabolous insects, orthologs of female-specific transformer (Tra), together with non-sex-specific auxiliary transformer-2 (Tra-2), promote female-specific splicing of doublesex (dsx) orthologs [18]. In the Drosophilidae family (but not other dipterans), sex-lethal (Sxl) controls Tra at a molecular level [19]. Outside of insects, Dmrts seem to act via sex-specific transcription. In *Caenorhabditis elegans*, there are many Dmrt paralogs. While several of them control sexual differentiation of specific tissues, none of them are regulated at the level of alternative splicing, but rather through sex-specific transcription. While putative Sxl and Tra-2 orthologs were identified in quite a few decapod species, in none of these cases were these genes reported to be differentially expressed or to exhibit gender-biased splice variance, in accordance with their lineage-specific link with sexual differentiation. Yet, the Dmrts are quite a phenomenal exception to this variability in sexual development mechanisms. It appears that Dmrts are the sole conserved entity across animalia in the context of sexual development [20]. While a range of Dmrts show broad tissue expression and are involved in regulating many developmental processes, some Dmrts have been identified as gender-specific or biased across Animalia, with a specific role in sexual development [3,20].

In just four cases across the animal kingdom, a Dmrt gene was indicated as the “master sex regulator”, whereby the gene was specifically found on a sex chromosome (either the female-specific W chromosome, or the male-specific Y chromosome) and, therefore, was directly associated with an ability to skew sexual development [3,7].

The Dmrts were recently shown to be a direct link between sex determination and sexual differentiation in decapods [21,22]. In two decapod species, it was reported that sex-biased Dmrts (Dmrt11E and Dmrt99B in the paleomonid *Macrobrachium rosenbergii* [21] and Dsx in the penaeid *Fenneropenaeus chinensis* [22]; all three Dmrts show higher expression in the testis than in the ovary) regulate the expression of the insulin-like androgenic gland hormone (IAG). The androgenic gland (AG) is an endocrine organ unique to male malacostracan crustaceans, which regulates the development and maintenance of the male gonad and additional sexual characteristics. The IAG is considered the single most conserved sexual differentiating factor across malacostraca [23]. Silencing Dmrts caused a significant reduction in IAG expression [21,22], suggesting that the IAG promoter is directly regulated by Dmrts. IAG production and secretion from the AG is additionally regulated by an eyestalk borne neuroendocrine complex known as the X-organ–sinus gland complex (XO–SG). The XO–SG governs a wide array of processes including molt, reproduction, and development. It was clearly shown that eyestalk ablation can induce molt, gonad maturation, and spawning and cause hypertrophy and hyperplasia of the AG [24,25,26,27,28,29]. These effects are attributed to a pleiotropic group of neuropeptides belonging to the crustacean hyperglycemic hormone (CHH) superfamily of neuropeptides [30,31,32]. The number of genes in this family is variable across decapod species, with some species having three and other species having as many as nine, according to many CHHs recently discovered through next-generation sequencing [9,15,33,34]. Some CHHs are involved primarily in sugar metabolism (like the CHH itself), while others mainly regulate molt (thus named the molt-inhibiting hormone; MIH) or gonad maturation (gonad-inhibiting hormone; GIH). The identity of the CHH receptors is heavily debated, with some evidence that they might be G-protein-coupled receptors (GPCRs), specifically those phylogenetically affiliated with a receptor of a derived hormone from insects [35,36]. Other lines of investigation led to the identification of several guanylyl cyclase as the CHH receptors. It is not yet clear if the Dmrts also regulate the CHHs.

Other than the IAG, additional insulin-like peptides (ILPs) were found in several decapod species [8,13]. The role of these ILPs in modulating IAG function or other processes is currently unresolved. The tyrosine kinase insulin receptor (TKIR) was identified as the key receptor for IAG in several decapod species [12,37], although TKIR activation by additional ILPs is yet to be assessed. Upon secretion, IAG is chaperoned in the hemolymph by an insulin-like growth factor-binding peptide (IGFBP) which is abundantly expressed across tissues [13,38]. IAG secretion is further facilitated by a membrane-anchored AG-specific factor (MAG), identified initially in the redclaw crayfish *Cherax quadricarinatus* [39] and later in the eastern spiny lobster *Sagmariasus verreauxi*, where a slight expression in the gonads was also detectable [11]. Modeling analysis concluded that the neural-derived ILPs share the same binding characteristic to IGFBP as IAG [8], suggesting they might also compete with IAG for receptor binding. Apart from the IAG receptor (TKIR), additional plausible ILP receptors were identified, including LGR3 and LGR101. These are two GPCRs which contain several leucine-rich repeat domains (LRR) in their N-terminus. The number of LRRs is indicative in the annotation of these receptors [40]. It is not yet clear whether these receptors and their putative ILP ligands are associated with the regulation of reproductive development in decapods.

The IAG mode of action is not entirely understood in decapod crustaceans. In the absence of the circulating IAG, the testes cease to function and even undergo degeneration [41,42]. It is also clear that testicular kinase activity changes rapidly upon AG [26] or IAG [12] exposure, although specific kinases have yet to be determined. In many decapod species, the removal of the AG from males or grafting of the AG into females causes a shift in the sexual characteristics, leading to feminization or masculinization [23]. In only one species to date, a full functional sex change was induced by IAG silencing [43] or injection of AG cells [44], utilizing genetic sex markers to trace the sex-changed individuals [45]. This has generated a very significant value in the ability to produce monosex populations of the commercially important giant freshwater prawn *M. rosenbergii* [43,44,46,47]. The freshwater prawn monosex aquaculture has opened many new opportunities for utilizing this genus in new niches, which extend beyond aquaculture and into disease management [48]. Spiny lobsters are widely distributed and have an appetite for echinoderms (which can imbalance marine ecosystems when proliferating unchecked). Spiny lobster monosex populations hold great potential if distributed in marine reserves outside their native origin, where they can manage the expanding echinoderm populations without the risk of long-term interruption, due to an inability to survive past their current generation, thus preventing the risk of introducing an invasive species. Such novel applications of monosex technology, alongside the continued expansion of aquaculture production, have motivated research into sexual development in decapods.

In the context of gonad maturation, expression of the major yolk protein encoding gene, vitellogenin, signifies a hallmark across animals. Once engaged in ripening, the female gonad accumulates vitellin, the major yolk protein. This is done via production of the precursor protein vitellogenin which undergoes post-translational processing, involving peptide cleavage and lipid sequestration. The main production site of vitellogenin in decapod crustaceans varies between species, with some producing the majority of their vitellogenin in the hepatopancreas before transport to the ovary via a vitellogenin receptor, while, in other species, the ovary itself is the main production site [49].

A putative factor associated with regulating sexual maturation in decapods is the gonad-stimulating hormone (GSH). Thought to be regulated by GIH, the elusive GSH is a presently undefined factor stemming from the central nervous system of decapods, potentially being one of three candidate neuropeptides: the red pigment concentrating hormone [50], whose cognate GPCR was recently deorphanized in the green shore crab *Carcinus maenas* [51], the glycoprotein alpha 2/beta 5 (GPA2/B5) dimer, whose receptor is an LRR containing GPCR, similar to the LGR3 and LGR101 receptors [6,33], or the crustacean female-specific hormone [52].

In this manuscript, we report a multi-tissue transcriptomic library of the ornate spiny lobster *Panulirus ornatus* and catalog all known genes related to sexual development in decapods. We identify a Y-linked Dmrt, namely, iDMY, the second sex-linked Dmrt recorded in invertebrates. This finding suggests conservation of the key master-sex regulator function of iDMY, with the eastern spiny lobster *S. verreauxi,* where iDMY was initially discovered, being somewhat of an anomaly considering the rarity of this mechanism across all studied animals. In addition, we take an exploratory approach to highlight a kinase specifically expressed in the gonads in these two spiny lobster species. The genes identified by this research will serve as a stepping stone toward developing sexual manipulation techniques for *P. ornatus*, an emerging species in closed-lifecycle aquaculture.

## 2. Materials and Methods

### 2.1. Sample Preparation and Sequencing

Six mature ornate spiny lobsters (*P. ornatus*; three males and three females, 1.37–1.83 kg) were purchased from wild-caught stocks captured in the Torres Strait in 2019 and reared at the University of the Sunshine Coast for at least one month prior to dissections. Six immature *P. ornatus* individuals (three males and three females, 300–350 g) derived from cultured stocks supplied by the Institute for Marine and Antarctic Studies (IMAS) aquaculture facility, Tasmania. Lobsters were reared as previously described [53]. Multiple tissues were dissected from 12 *P. ornatus* individuals, and samples were snap-frozen using liquid nitrogen and stored at −80 °C until use.

RNA was then extracted from up to 100 mg from each tissue using RNAZol (MRC, Melbourne, VIC, Australia), as previously described [54]. Total RNA was isolated from the testes (*n* = 6), ovaries (*n* = 6), and hepatopancreas (*n* = 11) of all individuals (with the omission of one immature male hepatopancreas). In addition, total RNA was extracted from the eyestalk, brain, thoracic ganglia, antennal gland, stomach, intestine, epidermal tissue, fat tissue, anterior gills, posterior gills, heart, and tail muscle of one mature *P. ornatus* male and one mature female (*n* = 2 per tissue). Total RNA was also isolated from three regions of the sperm duct (proximal, medial, and distal), as well as the hemocytes of one mature *P. ornatus* male and oviduct of one mature *P. ornatus* female (*n* = 1 per tissue). In total, RNA was extracted from 52 samples, covering multiple tissues, with multiple replicates for the gonads and hepatopancreas of immature and mature males and females.

RNA was quantified using NanoDrop 2000 (ThermoFisher, Scoresby, VIC, Australia) and tested using Bioanalyzer for integrity. At least 3 µg of clean and nondegraded RNA per sample were desiccated with RNAstable^®^ LD (Sigma-Aldrich, Brisbane, QLD, Australia) and sent to Novogene (Hong-Kong, China) for quality control, followed by library preparation (TrueSeq) and RNA Sequencing using the HiSeq2500 platform with paired-end 150 bp (PE150) sequencing. A minimum of 20 million reads were sequenced per sample.

### 2.2. Transcriptome Assembly and Quantification

Read trimming was conducted by Novogene on the basis of read quality (using unpublished in-house algorithms). Clean reads from four samples (immature and mature male testes and immature and mature female ovaries) were then de novo assembled using the CLC Genomics Workbench 8.0.3 (CLC; Qiagen, Chadstone, VIC, Australia) using automatic word size and bubble size to create the De Bruijn graphs, a minimum contig length of 200 bp with autodetection of paired distances, and scaffolding. Transcript expressions were quantified in each of the 12 gonad samples, relative to library size (calculated as reads per kilobase per million reads; RPKM) using CLC. The gonad-based de novo assembled transcriptome served as the reference file, and individual FASTQ gonad libraries were mapped to this reference under predefined parameters: no masking and random mapping, with similarity fraction 0.8.

### 2.3. Identification of Sex-Determination Pathway Orthologues

Analyses were conducted using NCBI Blast on the unfiltered transcriptome, using CLC and characterized sequences from *Caenorhabditis elegans* (Her-1 (NP_001024310.1), *TRA-1* (AAB59181.1), and TRA-2 (P34709.1)) and *D. melanogaster* (*Sxl* (NM_001169218.2), *TRA* (NM_079390.3), and *TRA-2* (NM_057416.3)). For *Dmrt* identification, orthologs from crustacean species were used, namely, *M. rosenbergii*-*Dsx11E* (KC801044), *Dsx99B* (KC801045), *E. sinensis*-*Dmrt-like* (HM051384); *Daphnia magna*-*Dsx11E* (BAG12871.1), *Dsx99B* (BAG12873.1), *Dsx93B* (BAG12872.1), *Dsx1α* (BAJ78307.1), *Dsx1β* (BAJ78308.1), and *Dsx2* (BAJ78309.1). Target hits were then computationally translated using the ExPASy Proteomic Server (http://web.expasy.org/translate/). The deduced amino-acid sequences were analyzed further using SMART (http://smart.embl.de/) to predict domain architecture and NCBI Blast (http://www.ncbi.nlm.nih.gov/) to assess conservation with characterized orthologues. All identified sequences and BlastP results (NCBI nr database) are available in Appendix A.

### 2.4. Phylogenetic Analyses of Dmrt Orthologues

Phylogenetic analyses were conducted with the complete amino-acid sequences of a range of characterized Dmrt peptides from other arthropods. The sequences were aligned using CLC Workbench (7.5.1) and a neighbor-joining tree was constructed; bootstrap analyses of 1000 replicates were carried out to determine confidence of branch positions. Clearly resolved clusters included the Dmrt11E, iDMY, iDmrt, and Dmrt99B orthologs. The DSX ortholog required separate alignment of the DNA-binding domain and oligomerization domain to bestow confidence in ortholog assignment. *P. ornatus* Dmrt sequences are available in Appendix A. Phylogeny, ortholog assignment, and alignments are available in Appendix A.

### 2.5. Mapping Analyses

In order to determine evidence of sex-specific splicing in the *Sv-Sxl* orthologues, mapping was conducted. Unassembled read libraries (stored as FASTQ files) of the brain, eyestalk, antennal gland, and gonad were combined, generating sex-specific male and female unassembled FASTQ read libraries. These two libraries were then mapped (following the parameters previously described) to each of the identified transcripts of *Sv-Sxl1* and *Sv-Sxl2*, to demonstrate any differential coverage between the sexes.

### 2.6. Differential Expression Analyses (DEA)

The mapped gonad libraries served as an experiment set at CLC where unpaired comparison was conducted among the immature and mature male testes and immature and mature female ovaries set as four groups. The resulting quantified transcripts were then assessed using proportion-based statistical analysis using Baggerley’s beta-binomial test with Bonferroni and FDR-corrected *p*-values (in CLC). The list of differentially expressed genes (corrected *p*-value ≤ 0.05) was then filtered in order to reduce noise in subsequent expression analyses. Filtering was conducted by removing transcripts with a difference between groups lower than 5 RPKM and fold change between groups lower than 10. Transcripts were considered as “specific” when the fold change was at least 10 with no more than 0.1 RPKM in one group.

### 2.7. Annotation

All resulting transcripts were submitted to ORFpredictor (http://proteomics.ysu.edu/tools/OrfPredictor.html). The predicted open reading frames were then annotated by BLAST search on the NCBI server, where only annotations with an E-value <1.00 × 10^−50^ were accepted. As a secondary method, the 27 specific ORFs also underwent a Pfam domain search of the “top 100 most common domains”, implementing an E-value cut-off of <1.00 × 10^−50^ (using CLC).

For a more in-depth analysis of the gonads, the gonad libraries were compared to the combined libraries of all the other 12 tissues following the same DEA criteria, generating a list of those transcripts that were upregulated in and specific to the gonads. Secondary DEAs were then run on this list comparing expression between the testis and ovary to highlight those transcripts that were specific to or upregulated in either testes or ovaries.

### 2.8. PCR Using Genomic DNA

Genomic DNA extraction from pleopods of 10 males and 10 females, followed by PCR, was conducted as previously described [7,13]. Primers were designed using Primer 3 (http://bioinfo.ut.ee/primer3-0.4.0/) and synthesized by IDT (Boronia, Victoria, Australia). The primers sequences are provided in Table 1. Amplicons were then electrophoresed alongside a 100 bp DNA ladder (Axygen, Wembley, WA, Australia) on a 1.2% agarose gel stained with ethidium bromide and visualized under ultraviolet (UV) light.

### 2.9. Sagmariasus Verreauxi Sample Collection and Quantitative RT-PCR Validation

Samples were collected from eight male and six female *Sagmariasus verreauxi* individuals (all individuals underwent primary sexual differentiation characterized by the presence of sex-specific gonopores but were not yet reproductively capable) that were cultured at the IMAS aquaculture facility as previously described. Samples were processed as previously described. Total RNA was isolated from male and female brain (BR), eyestalk (ES), antennal gland (AnG), testis (TS), and ovary (OV) using Trizol Reagent (Invitrogen, ThermoFisher), according to the manufacturer’s instructions. The complementary DNA (cDNA) was prepared by reverse-transcriptase reaction containing 1 μg of total RNA, using the Tetro cDNA Synthesis Kit (Bioline, Eveleigh, NSW, Australia) following the manufacturer’s instructions. Real-time qPCR was performed as previously described [54]. In brief, the cDNA served as a template for real-time qPCR using primers designed at the Assay Design Center (Roche website); see Table 1. Primers were mixed with the cDNA, FastStart Universal Probe Master (Rox; Roche Diagnostics, Indianapolis, IN, USA), and specific Universal ProbeLibrary Probe (Roche), and reactions were performed in Rotor-Gene Q (Qiagen). *Sv-18S* (GenBank accession no. KF828103) served to normalize quantification, which was calculated by equilibrating to the level of *Sv-18S* per sample and against the sample with the lowest value (2^−∆∆CT^).

## 3. Results and Discussion

### 3.1. Transcriptome Sequencing and Assembly

A total of 52 RNA-Seq libraries of *P. ornatus* were sequenced using Illumina HiSeq2500 PE150, with a minimum of 20 million reads per library. Following quality-based trimming, all libraries included a minimum of 98.7% clean reads. Four libraries, including immature and mature male testes and immature and mature female ovaries, were de novo assembled using CLC Workbench resulting in 170,063 transcripts.

### 3.2. Identification of Sexual Development Factors

Sexual development factors identified in closely related species served as queries for tBLASTn search of the newly generated tissue transcriptome, as well as the previously generated metamorphic stage transcriptome [55]. Key factors known to be related to the sexual development pathway (defined in Section 1) were clearly identified in *P. ornatus*. The sequences of all identified factors (with BlastP best hits, E-values, and identity and similarity percentages) are provided in Appendix A. A phylogeny of Dmrts is presented in Appendix A. The digital expression pattern for these factors across the 52 libraries is presented in Figure 1.

### 3.3. A Potential Sex-Determining Factor—The iDMY Gene Is Present Only in the Male Genome

In the screening of factors putatively involved in the sex determination pathway of *P. ornatus*, *Sxl* and *Tra-2* isoform-encoding transcripts showed broad tissue expression, with no clear bias between males and females. Among the Dmrts, *DSX* showed the highest and broadest expression followed by *Dmrt11E*. *Dmrt99B* was primarily expressed in the gonad, whereas *iDmrt* and *iDMY* were expressed across many tissues.

From the transcriptomic analysis, it is evident that the *P. ornatus iDMY* is expressed strictly in males (Figure 1). Genomic DNA extracted from males and females served as a template to amplify *iDmrt* and *iDMY*, showing that *iDmrt* can be amplified in both males and females, while *iDMY* is amplified strictly in the males (Figure 2). This result is in clear alignment with the male specificity of *iDMY* in *S. verreauxi* [7]. This highly conserved male-specific Dmrt (shared between two species from distantly related species of the infra-order Achelata) warrants further investigation as to its plausible role as a master sex regulator across Achelata, plausibly a conserved mechanism that is in striking contrast with the high variability in primary signals of sex determination genetic cascades observed across the animal kingdom.

### 3.4. Expression of the CHH Family of Neuropeptides

Most *CHH* family encoding transcripts are expressed primarily in the eyestalk as expected, except for *CHH-l-1* which is broadly expressed, with higher expression in the brain and thoracic ganglia than in the eyestalk. The *CHH* receptors show broad tissue expression with *CHHR2* expression primarily in the hepatopancreas and somewhat in the gonads. This result is concurrent with a formerly annotated *CHHR2* (Sv-GPCR_A12) in *S. verreauxi* [35]. Sex-biased expression was not observed in any of the *P. ornatus* CHHs or putative CHHRs.

### 3.5. Expression of the Insulin Endocrine Pathway

The insulin endocrine factors include IAG, which, as expected, shows highest expression in the distal region of the sperm duct, where the AG is situated in Achelata [14], as in many other decapod species [23]. While IAG is a very well-established sexual differentiating factor, leading to masculinization across malacostracans [23], its effect is not strictly conserved across the diverse decapods. In several cases, it has been shown that IAG is not expressed strictly in males or the AG [56]. In the blue swimmer crab, *Callinectes sapidus*, *IAG* expression was observed in the ovary [56] and hepatopancreas [57], suggesting involvement in ovarian development and carbohydrate metabolism. Apart from the distal part of the sperm duct, *IAG* expression is evident across tissues in both male and female *P. ornatus*, including the testis, ovary, male and female hepatopancreas, midgut, hindgut, and fat tissue (Figure 1). Further research is required to establish the reason behind this broad expression pattern. Given that IAG was indeed linked with growth regulation [42], perhaps this broad expression pattern positions IAG at the intersection of metabolism and reproduction.

In addition to *IAG*, *MAG* is also broadly expressed. Given that MAG was found to directly interact with IAG [39], this broad expression further supports a more diverse role for IAG in reproduction and metabolism. Interestingly, *MAG* expression overlaps with that of *ILP1*. Since ILP1, ILP2, and IAG were shown to share the same binding mechanism with the broadly expressed IGFBP [8], this suggests that the ILPs may also interact with MAG. In contrast with *ILP1* and *IAG*, *ILP2* shows a very restricted expression pattern with male-biased expression in the central nervous tissues and proximal sperm duct.

*TKIR* is expressed across all tissues with a male bias in three tissues. Low expression was observed in the male eyestalk, hindgut, muscle, and fat tissue, with no expression in the female equivalents. The broad expression is somewhat contradictory to previous work in eastern spiny lobster and freshwater prawn, where fewer tissues were examined, suggesting a more limited expression pattern. The eyestalk in crustaceans is a major source of inhibitory peptides, one of which inhibits AG proliferation and IAG production and secretion. Specific expression of *TKIR* in the male eyestalk could, therefore, act as an inhibitory feedback loop, which could prevent secretion of GIH, perhaps explaining in part the development of the AG specifically in males. In *D. melanogaster*, it was shown that certain muscle fibers develop only in males (the muscle of Lawrence required in males for mating movements). Different muscle groups are yet to be defined in spiny lobsters, which could perhaps explain the *TKIR* specific expression in *P. ornatus* male muscle. Having said this, TKIR is not male-specific and a broad expression is found across female tissues (Figure 1). Considering recent discoveries of additional ILPs in crustaceans [13] and IAG specific expression in males (at least in some species), TKIR may well be functioning as a receptor for ILP1 and ILP2, as well as IAG.

### 3.6. Insulin-Like Peptide Receptors Analysis

In the vertebrates, relaxin and insulin-like peptide 3 (INSL3) are known to function through two relaxin receptors (RXFP1 and RXFP2, respectively) from the G-protein-coupled receptor (GPCR) family (Type C1) [58,59]. Homologs of these receptors have been identified in *D. melanogaster* (dLGR4 and dLGR3 respectively) and many other arthropods, including *S. verreauxi* [35] and other decapods [33]. On closer inspection, the homologs of dLGR3s appear to be well conserved in the decapods, identified in all eight species examined [10,33]. Analysis in *S. verreauxi* shows the LGR3 homolog to be gonad specific, with a strong female bias (Figure 3A). This strong ovarian expression is also apparent in *P. ornatus*, with increased expression seen in the mature ovarian tissue (Figure 1). However, the more comprehensive transcriptional profile in *P. ornatus* also reveals expression in the neuroendocrine tissues of both males and females.

In contrast, the classical dLGR4 homologs were not identified in five of the eight decapod species, *S. verreauxi* being one. We were able, however, to identify a nonclassical relaxin receptor in *S. verreauxi* and *P. ornatus*, homologous to dLGR4 in the 7 transmembrane domain (7TM). This receptor displays an extended ectodomain, containing 12 low-density lipoprotein repeats (LDLa), as opposed to the one LDLa common to the classical relaxin receptor family [59] (Figure 3C,D). This nonclassical class of GPCRs (LGR101s) has been defined as the Type C2 class [59] and appears to be limited to the more ancient groups (Figure 4; green box). Expression of *LGR101* in *S. verreauxi* is found in the neuroendocrine tissues (brain and eyestalk), with greatest expression in the female brain (Figure 3B). A similar pattern is seen in *P. ornatus*, where it is found in the brain, thoracic ganglia, and, to a lesser extent, the eyestalk, with additional expression in the ovary and array of somatic tissues not analyzed in *S. verreauxi* (Figure 1). The conserved expression of the *LRG101*s in the neuroendocrine tissues in Achelata, is similar to that of the mollusk, where the receptor was first characterized in the central nervous system of *Lymnaea stagnalis* [60]. Considering its phylogeny, we suggest that LGR101s may be a subclass of the classical dLGR4 relaxin receptor.

Of these receptors, dLGR3 has been deorphanized in *D. melanogaster*, shown to mediate the growth-coordinating effects of Dilp8 [61,62,63]. It has been suggested that the decapod IAGs are the lineage-specific homologs of Dilp8 and, therefore, IAG signaling occurs through the dLGR3 homologs [33]. This work also suggests that the “insulin-like” ILPs function through the TKIR. However, considering the IAG activation [12] and the phenotypic effects [37] of the TKIR in the male sexual differentiation, we still advocate that the TKIR is the primary receptor for IAG. This is further strengthened by the fact that we subsequently identified a distinct ILP subclass across decapods, the ILP2s, which show far closer homology to Dilp8 than the IAGs [8]. Expression of *ILP2* in *S. verreauxi*, *C. quadricarinatus* [8], and *P. ornatus* is limited to the neuroendocrine tissues (brain, eyestalk, and thoracic ganglia), and, in light of the deeper tissue analysis in *P. ornatus* (compared with *S. verreauxi*), expression of the *LGR3* homolog is also apparent across these tissues. A more comprehensive analysis in *S. verreauxi* may discern a similar pattern. We, therefore, hypothesize that the ILP2 class is more likely to function through the decapod LGR3 homolog. However, a multi-ligand, multi-receptor assay is required to validate these predictions.

Furthermore, although we propose particular ligand–receptor interactions, crosstalk is a common feature of insulin signaling systems [58,64]. In the context of IAG and sexual development, it is likely that the hormone also shows functionality through these relaxin GPCRs. Indeed, work in *M. rosenbergii* provides such evidence; silencing of IAG can induce complete sex change, causing a genetic male to sexually differentiate and mature as a functional female [43], whereas, in contrast, silencing the TKIR appears to advance male sexual differentiation (specifically the emergence of the sexually dimorphic appendix masculina) whilst impeding spermatogenesis, thus preventing sperm maturation [37]. This provides strong indication that more than one receptor mediates the integrated processes of sexual differentiation through to the gametogenesis to fulfill reproductive maturation. Similarly, it is probable that the TKIR holds functional potential for any of the two additional ILPs now identified in decapods. Therefore, we believe that the incorporation of these relaxin-like GPCRs offers the next step in understanding the broader sex-differentiating effects of IAG in the decapods.

### 3.7. Ovarian Maturation Factors

Gonad stimulatory factors were previously proposed in crustaceans to be red pigment concentrating hormone (RPCH), GPA2/B5, or crustacean female-specific hormone (CFSH) [6]. It is interesting to note that RPCH is a decapeptide related to the gonadotropin-releasing hormone (GnRH) superfamily, which governs gonad maturation in vertebrates. Downstream of the GnRH are the gonadotropins, which evolved on the basis of GPA2/B5 ancestral gene templates. It is tempting to speculate that these factors are indeed conserved in the context of gonad maturation from invertebrates to vertebrates. In the case of vertebrates, the downstream effector of sexual maturation is sex steroids, whereas the case for invertebrates is more nebulous. There is strong evidence for a parallel mechanism in arthropods, whereby ecdysteroid hormones are the key downstream effectors [65]. However, this is complicated by the lineage-specific emergence of the CHHs in crustaceans (produced in the XO–SG). In another parallel with vertebrates, it is yet to be elucidated whether the arthropod ILPs function as relaxins, while the GPA2/B5 receptor is from the same family of LRR GPCRs. With regard to the expression of these putative GSHs and their cognate receptors, *RPCH* and its receptor are strictly expressed in the neural tissues, while *GPB5* is also expressed in the ovary across species, whereas *GPA2* is not [5,9,66]. Since both *GPA2* and *GPB5* are coexpressed in the neural tissues, they could potentially dimerize and then act on the gonad, whereas, in the gonad itself, only *GPB5* is expressed, which could regulate the function of dimerized factors from the CNS. The *LGR1*, activated by GPA2/B5, is expressed in the neural tissues, as well as the testes, sperm duct, and immature ovaries, and some somatic tissues. While there is a high expression of *GPB5* in mature ovaries, expression of *LGR1* is not apparent (Figure 1). Given the expression pattern of *LGR3* and *LGR101*, as well as their close counterpart *LGR1*, it could be suggested that ILP1, ILP2, GPB5, and possibly a GPA2/B5 dimer all compete for binding to these LRR-containing GPCRs to regulate gonad maturation. The CFSH is expressed in the eyestalk and testes, suggesting it does not regulate ovarian maturation. It is not clear what receptor is activated by CFSH.

### 3.8. Identification of Gonad-Specific Expressions

When exploring which transcripts were upregulated in the four gonad groups, each with three replicate libraries (with a minimum of 10× fold change between groups), 2129 transcripts were upregulated in the ovaries, of which 270 transcripts were ovary-specific, and 1206 transcripts were upregulated in the testes, of which 485 were testis-specific. The immature versus mature testis samples showed 177 versus 55 enriched transcripts, respectively. The immature versus mature ovary samples showed 526 versus 112 enriched transcripts, respectively (Figure 5). Many of the transcripts were noncoding or incomplete. Interestingly, quite a few gonad-enriched sequences were kinases. Assessing the paralogs of these kinases for gonad specificity in *S. verreauxi* identified a single kinase which is gonad-specific in both species (Figure 1 and Appendix A). Considering the key function of IAG in regulating testis activity, it is clear that its function would be mediated via amplified cascades of kinases as previously mentioned.

### 3.9. Concluding Remarks

In summary, all the factors known in the sexual development pathway in decapod crustaceans were identified, and their expression across tissues was digitally calculated (Figure 1). The male-specific iDMY is conserved between the eastern and ornate spiny lobsters (*S. verreauxi* and *P. ornatus*), indicating a conserved central function in sex determination. The predicted sexual development system in decapods potentially parallels the hypothalamus–pituitary gland–gonad (HPG) neuroendocrine axis in vertebrates, where GnRH secreted from the hypothalamus stimulates the release of gonadotropins from the pituitary gland, which in turn stimulates steroid synthesis and release by the gonads. In decapod crustaceans, the XO–SG inhibits the release of gonadotropins from the central nervous system (with clear links between the GnRH and RPCH decapeptides and the gonadotropins and the GPA2/B5 glycoproteins). The decapod equivalent of vertebrate sex steroids, secreted by the gonads in response to the gonadotropin stimuli, is yet to be elucidated, with ecdysteroids suggested as candidate parallels. The gonad-specific kinase identified in this study in two spiny lobster species is a potential downstream effector of IAG, joining multiple other factors identified as candidates for further research to better understand sexual development in decapods. Figure 6 provides a schematic depiction of these conclusions.

## Figures and Tables

**Figure 1 genes-11-01150-f001:**
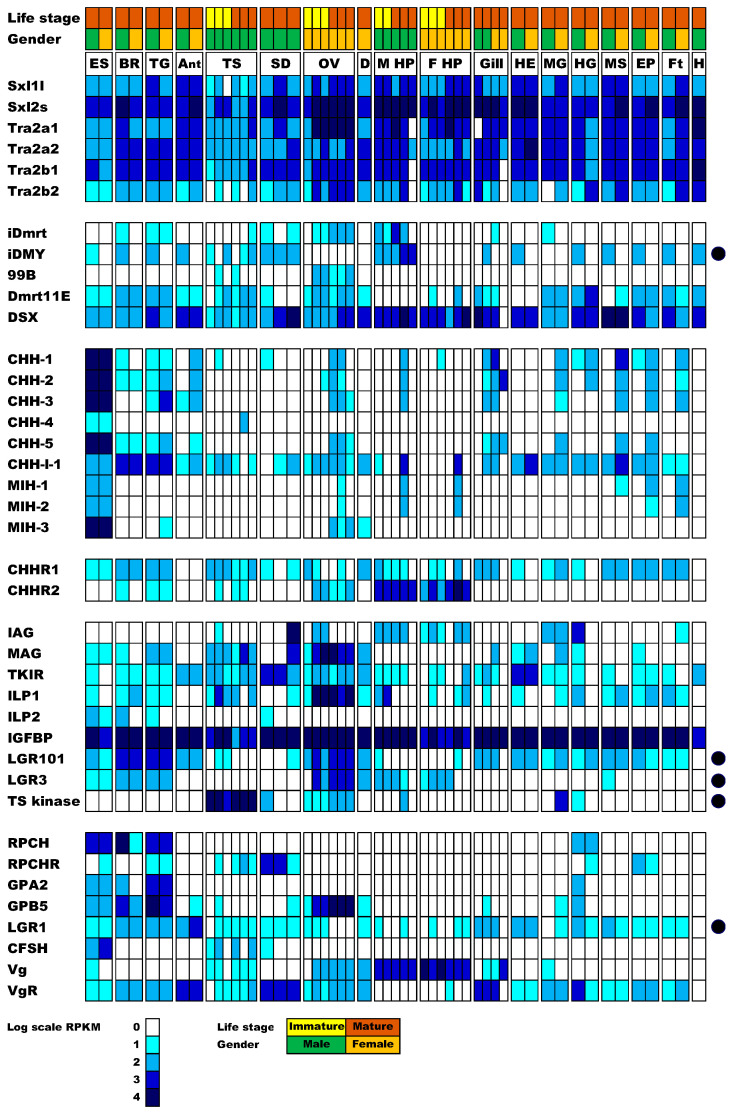
Expression of key sexual development transcripts across multiple tissues on the ornate spiny lobster *Panulirus ornatus*. The expression of key transcripts involved in sexual development was calculated as reads per kilobase per million reads (RPKM) in RNA-Seq libraries of multiple *P. ornatus* tissues. Log-transformed expression is represented by gradient color shading. Tissues from left to right: ES—eyestalk (male, female); BR—brain (male, female); TG—thoracic ganglia (male, female); AnG—antennal gland (male, female); TS—testis (3 × immature, 3 × mature males); SD—sperm duct (proximal, medial, distal); OV—ovary (3 × immature, 3 × mature females); D—oviduct; M HP—male hepatopancreas (2 × immature, 3 × mature males); F HP—female hepatopancreas (3 × immature, 3 × mature females); Gill—anterior and posterior of male, then female; HE—heart (male, female); MG—midgut (male, female); HG—hindgut (male, female); MS—tail muscle (male, female); EP—epidermis (male, female); Ft—fat tissue (male, female); H—male-derived hemocytes. Dots on the right-hand side signify transcripts further analyzed in this research.

**Figure 2 genes-11-01150-f002:**
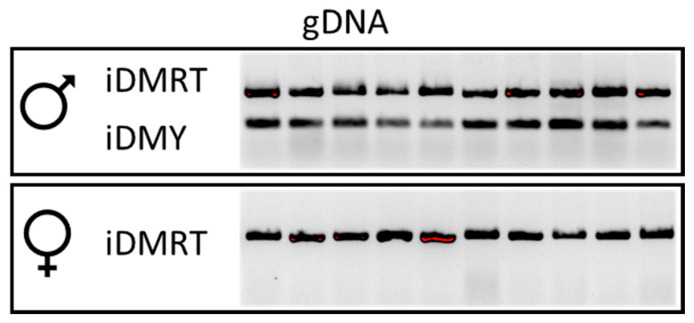
*DMY* is specifically found in the male genome. Using genomic DNA from males and females as a template (*n* = 10 per sex), PCR successfully amplified *iDmrt* and *iDMY*. A single band of expected sizes is amplified per gene, with *iDmrt* amplified in both males and females while *iDMY* is amplified strictly in males.

**Figure 3 genes-11-01150-f003:**
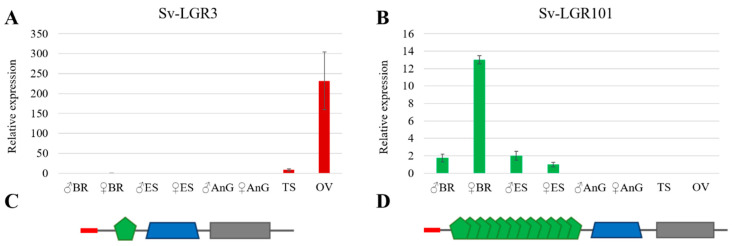
Expression and domain structure of leucine-rich repeat (LRR) G-protein-coupled receptor (GPCR) Sv-LGR3 and Sv-LGR101 in *Sagmariasus verreauxi*. Sv-LGR3 (**A**) and Sv-LGR101 (**B**) real-time qPCR expression analysis from *S. verreauxi* tissue from male and female brain (BR), eyestalk (ES), antennal gland (AnG), testis (TS), and ovary (OV). Data represent the mean ± standard error of the mean (SEM) (*n* = 8). Relaxin-like LRR GPCR domain structure of Sv-LGR3 (**C**) and Sv-LGR101 (**D**). The red box denotes the signal peptide, green pentagons denote the low-density lipoprotein (LDLa) extracellular repeats, the blue trapezium denotes the seven-domain transmembrane region, and the grey box represents the intracellular domain. LGR3 is a classic relaxin-type GPCR (Type C1) with one LDLa domain in the extracellular region. LGR101 is a nonclassical relaxin-type GPCR (Type C2) due to the 12 LDLa repeats in the extracellular region, making it part of the LGR101 family.

**Figure 4 genes-11-01150-f004:**
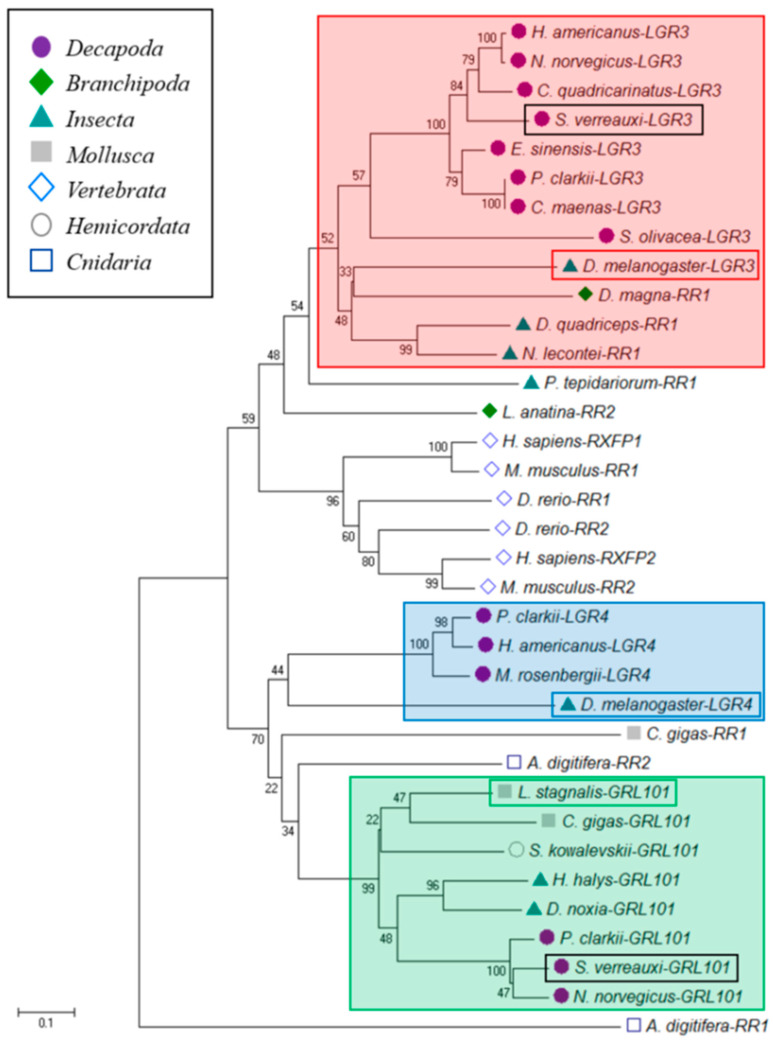
Neighbor-joining phylogram of the 7 transmembrane domains (7TMs) of dLGR3, dLGR4, and GRL101 homologs identified in eight decapod species; bootstrap values are shown at each node and were performed with 1000 replicates. The scale bar indicates number of amino-acid substitutions per site and key highlights the taxonomy. dLGR3 is boxed in red, dLGR4 is boxed in blue, and LGR101 is boxed in green, with all of the model species emphasized in boxes of corresponding colors. *S. verreauxi* relaxin receptor 1_LGR3 (Sv-RR1, Accession number KY427011) and G-protein coupled-receptor 101 (Sv-LGR101, Accession number KY427010) are boxed in black.

**Figure 5 genes-11-01150-f005:**
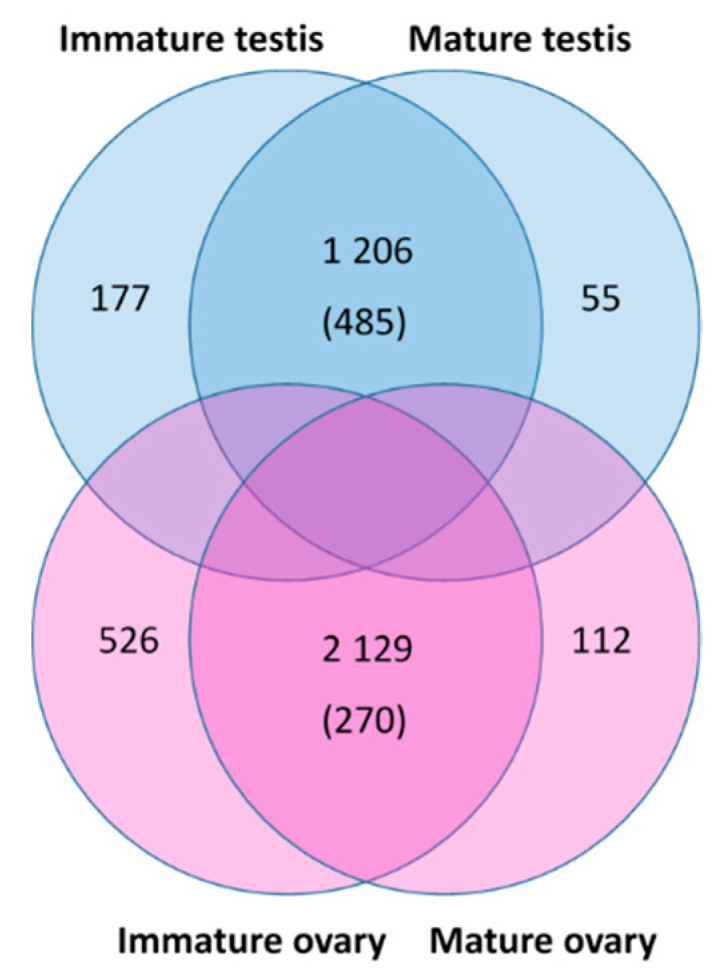
Venn diagram of differentially expressed genes between testis and ovary samples of different life stages. Comparison of immature and mature testis samples (blue) identified 177 differentially expressed genes (DEGs) enriched in the immature testis and 55 DEGs enriched in the mature testis samples. Comparison of immature and mature ovary samples (pink) identified 526 DEGs enriched in the immature ovary and 112 DEGs enriched in the mature ovary samples. Comparison of testis and ovary samples identified 1206 DEGs enriched in the testis (of which 485 transcripts are testis-specific) and 2129 DEGs enriched in the ovary (of which 270 transcripts are ovary-specific).

**Figure 6 genes-11-01150-f006:**
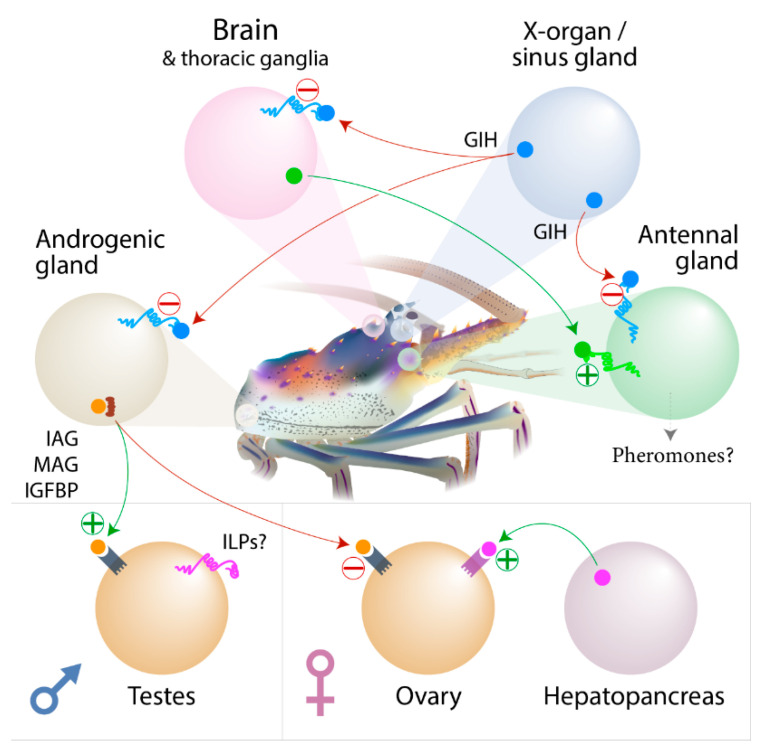
Proposed model for decapod sexual development molecular interactions. The predicted sexual development system in decapod crustaceans involves the gonad inhibiting hormone (GIH—blue), which is secreted from the x-organ–sinus gland complex and inhibits the release of gonadotropins from the central nervous system (brain and thoracic ganglia). These gonadotropins (plausibly red pigment concentrating hormone (RPCH), GPA2/B5, or crustacean female-specific hormone (CFSH)—green) then bind receptors on target cells to stimulate reproductive capacity in the gonads and potentially also the antennal gland (where they potentially stimulate pheromones production and release). The GIH potentially inhibits this directly. The decapod equivalent of vertebrate sex steroids, secreted by the gonads in response to the gonadotropin stimuli, is yet to be elucidated, with ecdysteroids suggested as candidate parallels. The gonad-specific kinase we identified in two spiny lobster species is a potential downstream effector of IAG, joining multiple other factors identified as candidates for further research to better understand sexual development in decapods. The male-specific iDMY is conserved between the eastern and ornate spiny lobsters (*S. verreauxi* and *P. ornatus*), indicating a conserved central function in sex determination.

**Table 1 genes-11-01150-t001:** Primers used in this study.

Primer Name	Gene/Amplicon Size (nt)	Primer Sequence 5′–3′; Accession Number
*Po-iDMY F*	*Po-iDMY/322nt*	ACACACTTAAGCCGTCTCCA
*Po-iDMY R*		TTTCATAACGCCGTGGTTCC
*Po-iDmrt F*	*Po-iDmrt/528nt*	GCAGCCTGAATATGAGGGGT
*Po-iDmrt R*		AGTAAGGCAAGTTGACGGGA
*qSv-LGR3 F*	*Sv-LGR3/73nt*	GACGGAGTGTCATCGTTCG
*qSv-LGR3 R*		CCACAATCACCAGCCACA Accession number: KY427011 (Sv-RR1)
*qSv-LGR101 F*	*Sv-LGR101/61nt*	GACATCGTGGCTGTGTCG
*qSv-LGR101 R*		GCTGGACATTCCAACGACTT Accession number: KY427010 (Sv-GRL101)

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
