# Peer review of "Multi-Tissue Transcriptome Analysis Identifies Key Sexual Development-Related Genes of the Ornate Spiny Lobster (Panulirus ornatus)"

_genes, 2020, doi:10.3390/genes11101150_

Round 1

Reviewer 1 Report

Ventura et. al provide a multi-tissue transcriptomic library of the spiny lobster along with 39 genes linked to decapod sex development. The authors found a previously identified (by the authors) Y-linked Dmrt, which was found to be the master sex regulator in another spiny lobster (with ~167 MY divergence). The authors propose a decapod equivalent HPG neuroendocrine axis in vertebrates, where a newly identified gonad specific kinase plays potential downstream effector role of insulin-like androgenic gland hormone. This study contributes to the overall knowledge of sex development in spiny lobsters as well as invertebrates and identifies key areas of future study related sexual development in decapods including the potential role of ecdysteroids, gonad specific kinase and iDMY. I find the story to be interesting and easy to follow. I have several suggestions/ criticisms:

  • Small self-citation issue. The first author has 24 publications in the reference list, and while I recognize this niche field is largely dominated by the authors, I see at least 2 sentences in the introduction that I think should be cited by other sources. For example, the first sentence (lines 35-37) is a broad statement that has been written again and again in sex chromosome/differentiation papers much earlier than the author’s own paper published in 2018. Likewise, the second sentence (lines 37-39) could cite any number of sources other than the author’s own 2018 paper. I also recognize that the author may still include those sources in their reference list, as they are cited for other parts of the article.
  • There is no citation for information presented in lines 60-70 related to Dmrt, Tra-2 and Sxl in the fruit fly.
  • Mild punctuation issue: In multiple places, I notice the authors do not have a comma after a longer than 3 word introductory phrase. I think it’s important for overall clarity of the paper to include these commas. 3 examples include: line 116- comma needed after “IAG”; line 118 comma needed after “species”; line 282- comma needed after “analysis”
  • Just curious, have the authors looked at ATRX? It has been shown to be involved in sex differentiation in species from mammals to amphibians.
  • This study decidedly eliminates any novel findings for genes that may play a role in this particular spiny lobster sex differentiation by not doing a couple of basic things I’ve attempted to highlight here: 1) In figure 5, the venn diagram shows many differentially expressed transcripts that are potentially specific to female vs male, but these transcripts are never identified. A quick blast search could help elucidate what some of these transcripts and perhaps provide more information to the authors. 2) authors might also consider comparing all data from males vs females by mapping reads back to a transcriptome assembly (built using all of the reads) and looking for transcripts that are present in one sex vs the other. I recognize that this study does not absolutely need this to be done in order to be published, but I do think identifying those differentially expressed transcripts from the venn diagram in a supplementary file would be useful to readers

Author Response

Responses to Reviewer 1 comments

  • Small self-citation issue. The first author has 24 publications in the reference list, and while I recognize this niche field is largely dominated by the authors, I see at least 2 sentences in the introduction that I think should be cited by other sources. For example, the first sentence (lines 35-37) is a broad statement that has been written again and again in sex chromosome/differentiation papers much earlier than the author’s own paper published in 2018. Likewise, the second sentence (lines 37-39) could cite any number of sources other than the author’s own 2018 paper. I also recognize that the author may still include those sources in their reference list, as they are cited for other parts of the article.

Thank you for this valuable comment. We added additional references to these two sentences with earlier findings. We chose to keep the reference to our work which is more relevant for crustaceans though.

Note for the editor – these citations were added below the reference list at the bottom of the document. They should be embedded into the reference list and the numbers should be edited accordingly.

  • There is no citation for information presented in lines 60-70 related to Dmrt, Tra-2 and Sxl in the fruit fly.

Three relevant citations were added. Note for the editor – these citations were added below the reference list at the bottom of the document. They should be embedded into the reference list and the numbers should be edited accordingly.

  • Mild punctuation issue: In multiple places, I notice the authors do not have a comma after a longer than 3 word introductory phrase. I think it’s important for overall clarity of the paper to include these commas. 3 examples include: line 116- comma needed after “IAG”; line 118 comma needed after “species”; line 282- comma needed after “analysis”

Changed as suggested.

  • Just curious, have the authors looked at ATRX? It has been shown to be involved in sex differentiation in species from mammals to amphibians.

We have not investigated ATRX. This resource (of multiple tissues with biological replicates) offers an extensive resource for further exploratory analyses. To facilitate this, we are developing a freely available, web-based interface to access this data, in the hope that the wider scientific community can apply our transcriptomic data to their own area of interest. The interface will provide a simple online tool, allowing researchers to BLAST any gene of interest to visualise its expression pattern across all tissues of our transcriptome.

  • This study decidedly eliminates any novel findings for genes that may play a role in this particular spiny lobster sex differentiation by not doing a couple of basic things I’ve attempted to highlight here: 1) In figure 5, the venn diagram shows many differentially expressed transcripts that are potentially specific to female vs male, but these transcripts are never identified. A quick blast search could help elucidate what some of these transcripts and perhaps provide more information to the authors. 2) authors might also consider comparing all data from males vs females by mapping reads back to a transcriptome assembly (built using all of the reads) and looking for transcripts that are present in one sex vs the other. I recognize that this study does not absolutely need this to be done in order to be published, but I do think identifying those differentially expressed transcripts from the venn diagram in a supplementary file would be useful to readers

The transcripts identified as DEGs in figure 5 were subjected to BLAST searches and screened for genes of potential interest, however these data are not reported in the manuscript as there were no findings of interest. In addition, these transcripts were cross-compared with the previously published eastern spiny lobster database to identify DEGs present in both species, this is how we identified the testis-specific kinase. For this reason, we decided not to include a supplementary file of all DEGs in P. ornatus to avoid drawing the readers’ attention to genes that do not share biological function in sexual development across the two species. Further research could indeed highlight additional factors of interest by cross-comparisons across species.

Reviewer 2 Report

In this study, to get insight on the genetic basis of sexual development in the ornate spiny lobster, Ventura et al. performed RNA-seq analyses using various tissues from males and females. This study has the potential for understanding the molecular mechanism and evolution of sexual development in crustaceans in addition to the development of monosex culture in economically important decapod species. However, this reviewer had the two major concerns mainly related to experimental design and data. This preliminary state of experimental data must be improved before publication.

1) For the transcriptomic analyses, multiple replicates were used for gonads and hepatopancreas as described on page 4, line 175. However, the other tissues including eyestalk and brain were analyzed with two or one replicates despite their important roles for sexual development. More importantly, there was no statistical analysis in this transcriptome data that occupies the main part of this manuscript.

2) There was no information on how orthologs in Panulirus are conserved at amino acid sequence levels. E-values and scores are necessary together with data written in the supplementary file. For more important genes such as DM domain genes, amino acid sequence alignment and phylogenetic analysis are required to confirm whether they are bonafide orthologs or not. For example, how did the author conclude that this prawn has DSX? Does this DSX have any motifs or phylogenetical relationships special for Dsx orthologs?

Minor comments

1) In the abstract, the content of experiments and their results was described only with one sentence. Background information needs to be reduced.

2) In this study, target gene expressions were normalized with 18S ribosomal RNA expression. In Figure 3, values for expression levels of several genes show more than one. This is strange. More explanation about a normalization method is needed.

3) In figure 3, no label.

4) The difference between non-classical and classical dLGR4 was unclear. What is the definition of “classical”?

Author Response

Responses to Review 2 comments

Major comments

  • For the transcriptomic analyses, multiple replicates were used for gonads and hepatopancreas as described on page 4, line 175. However, the other tissues including eyestalk and brain were analyzed with two or one replicates despite their important roles for sexual development. More importantly, there was no statistical analysis in this transcriptome data that occupies the main part of this manuscript.

Based on our prior knowledge across multiple decapod species (including the eastern spiny lobster, redclaw crayfish, European lobster, tiger prawn, whiteleg prawn, giant freshwater prawn, land crabs, swimmer crabs, mud crabs and mitten crabs), we chose to sequence replicates of those tissues (the gonads and hepatopancreas) that show differential expression associated with sexual development.

The only component that shows differential expression outside the gonads and hepatopancreas, is the crustacean female specific hormone (which is differentially expressed in the X-organ ganglia in the eyestalk, but only in brachyurans, not in spiny lobsters, clawed lobsters etc.). The only other exception is of course the well-studied IAG in the androgenic gland. This study therefore provides: 1) a comprehensive transcriptome of multiple tissues with two biological replicates (one male and one female) to enable future studies into multiple traits in lobsters; and 2) a focussed analysis of sexual development, hence the additional replicates of gonads and hepatopancreas. In doing so, this study reports a total of 52 RNA-Seq libraries, which is, to the best of our knowledge, far more than any single decapod study to date. Regarding statistical analysis, the identification and description of sexual development factors does not require statistics, as the aim of this work is to build on former studies, identifying novel genes and common pathways to provide a better molecular understanding of sexual development.  The highlights include the first identification of a genetic sex marker in a commercially important species, finding a conserved iDMY that is Y-linked in two spiny lobster species and overall an integrated description of the conserved features of sexual development across decapods.

  • There was no information on how orthologs in Panulirus are conserved at amino acid sequence levels. E-values and scores are necessary together with data written in the supplementary file. For more important genes such as DM domain genes, amino acid sequence alignment and phylogenetic analysis are required to confirm whether they are bonafide orthologs or not. For example, how did the author conclude that this prawn has DSX? Does this DSX have any motifs or phylogenetical relationships special for Dsx orthologs?

The exceptionally high similarity at the DNA level between those orthologs identified in the ornate lobster and previously studied eastern spiny lobster, is striking. This makes comparisons of the amino acid level somewhat redundant. We therefore chose to avoid including these analyses (that would highlight the obvious) so that the readers can focus their attention on the highlights.

 Minor comments

  • In the abstract, the content of experiments and their results was described only with one sentence. Background information needs to be reduced.

This is an authorship style preference. We chose to emphasize the background which laid the foundation for the identification of the key factors associated with sexual development in this study.

  • In this study, target gene expressions were normalized with 18S ribosomal RNA expression. In Figure 3, values for expression levels of several genes show more than one. This is strange. More explanation about a normalization method is needed.

The method used for normalizing the qPCR is 2-ddCt, which is well regarded as the convention for relative expression. This method has been practiced recursively in our group for many years and has been cited and explained in detail previously. The relative quantification output includes arbitrary units whereby the lowest expressing sample is normalized to 1 and all other samples are calculated relative to it.

  • 3) In figure 3, no label.

Can the reviewer please elaborate what they refer to?

  • 4) The difference between non-classical and classical dLGR4 was unclear. What is the definition of “classical”?

Classical refers to the conserved receptor which conforms to the common domain features of an LGR with one low-density lipoprotein repeat (LDLa). The non-classical defines a receptor that has a different ectodomain which contains twelve LDLa repeats, but is conserved in the 7 transmembrane domain.

This definition is described in full in text, lines 348-353: “We were able however, to identify a non-classical relaxin receptor in S. verreauxi and P. ornatus, homologous to dLGR4 in the 7 transmembrane domain (7TM) but displaying an extended ectodomain, containing twelve low-density lipoprotein repeats (LDLa), as opposed to the one LDLa common to the classical relaxin receptor family [54] (Figure 3C, D) . This non-classical class of GPCRs (LGR101s) has been defined as the Type C2 class [54] and appears to be limited to the more ancient groups (Figure 4; green box).”.

Round 2

Reviewer 2 Report

This reviewer agreed with the authors’ comment that this study focused more on identification (or presence) of potential regulators for sexual development. Therefore, the statistical calculation for the difference in gene expression was not so important. The other following reviewer’s concerns have not been addressed.

1) Because this paper focused more on the identification of orthologs, molecular evidence is necessary to explain the exceptionally high similarity at the DNA level between those orthologs. The authors did not show data on the similarity between orthologs, which prevented this reviewer from understanding the high similarity and whether the authors analyzed the bona fide orthologs. The authors also did not respond to this reviewer’s comments, “How did the author conclude that this prawn has DSX? Does this DSX have any motifs or phylogenetical relationships special for Dsx orthologs?” These concerns should be addressed.

2) Please read the instructions for authors in this journal (https://www.mdpi.com/journal/genes/instructions#preparation) and revise the abstract by following the instructions. 

3) In figure 3, this reviewer understood the authors’ method of 2-ddCt calculation but could not see any value “1” in the right panel.

4) In figure 3 legend, explanation of labels (A), (B), (C), and (D) was described, but in the figure its self, there was no label.

Author Response

Response to reviewer #2 (R2):

This reviewer agreed with the authors’ comment that this study focused more on identification (or presence) of potential regulators for sexual development. Therefore, the statistical calculation for the difference in gene expression was not so important. The other following reviewer’s concerns have not been addressed. 

1) Because this paper focused more on the identification of orthologs, molecular evidence is necessary to explain the exceptionally high similarity at the DNA level between those orthologs. The authors did not show data on the similarity between orthologs, which prevented this reviewer from understanding the high similarity and whether the authors analyzed the bona fide orthologs. The authors also did not respond to this reviewer’s comments, “How did the author conclude that this prawn has DSX? Does this DSX have any motifs or phylogenetical relationships special for Dsx orthologs?” These concerns should be addressed.

A supplementary file was added, detailing the Dmrts orthologs, including DSX domain architecture. 

2) Please read the instructions for authors in this journal (https://www.mdpi.com/journal/genes/instructions#preparation) and revise the abstract by following the instructions. 

The abstract is adhering to the instructions. It is less than 200 words, has background, methods, results and conclusions summarised, albeit with emphasis on the background.

3) In figure 3, this reviewer understood the authors’ method of 2-ddCt calculation but could not see any value “1” in the right panel.

The RQ values in the right panel (now labelled Fig. 3B) were not correct and are now modified.

4) In figure 3 legend, explanation of labels (A), (B), (C), and (D) was described, but in the figure its self, there was no label.

Labels were added to Figure 3.
